# Metabolic and Anti-Inflammatory Protective Properties of Human Enriched Serum following Artichoke Leaf Extract Absorption: Results from an Innovative Ex Vivo Clinical Trial

**DOI:** 10.3390/nu13082653

**Published:** 2021-07-30

**Authors:** Fabien Wauquier, Line Boutin-Wittrant, Aurélien Viret, Laure Guilhaudis, Hassan Oulyadi, Asma Bourafai-Aziez, Gwladys Charpentier, Guillaume Rousselot, Emmanuel Cassin, Stéphane Descamps, Véronique Roux, Nicolas Macian, Gisèle Pickering, Yohann Wittrant

**Affiliations:** 1Clinic’n’Cell SAS, Faculty of Medicine and Pharmacy, TSA 50400, 28 Place Henri Dunant, CEDEX 1, F-63001 Clermont-Ferrand, France; Fabien_Wauquier@gmx.fr (F.W.); linewittrant@gmail.com (L.B.-W.); 2Laboratoire COBRA (UMR 6014 & FR 3038), CNRS, INSA de Rouen, UNIROUEN, Normandie Université, F-76821 Mont-Saint-Aignan, France; aurelien.viret1@univ-rouen.fr (A.V.); Laure.Guilhaudis@univ-rouen.fr (L.G.); hassan.oulyadi@univ-rouen.fr (H.O.); 3EVEAR EXTRACTION, 48 Route de Gennes, LD Félines, CEDEX 4, F-49320 Coutures, France; a.bourafai@evear-extraction.com (A.B.-A.); g.charpentier@evear-extraction.com (G.C.); g.rousselot@evear-extraction.com (G.R.); e.cassin@evear-extraction.com (E.C.); 4CNRS, CHU Clermont Ferrand, ICCF, Clermont Auvergne INP, Université Clermont Auvergne, CEDEX 1, F-63000 Clermont-Ferrand, France; s_descamps@chu-clermontferrand.fr; 5CIC INSERM 1405/Plateforme d’Investigation Clinique CHU Gabriel Montpied, 58 Rue Montalembert, CEDEX 1, F-63000 Clermont-Ferrand, France; v_morel@chu-clermontferrand.fr (V.R.); nmacian@chu-clermontferrand.fr (N.M.); gisele.pickering@uca.fr (G.P.); 6INRAE, UMR 1019, UNH, CEDEX 1, F-63009 Clermont-Ferrand, France; 7UMR1019 of Human Nutrition, Clermont Auvergne University, BP 10448, CEDEX 1, F-63000 Clermont-Ferrand, France

**Keywords:** clinical trial, metabolic disorders, inflammation, artichoke, polyphenols, metabolites, NMR, human cells, hepatocytes, chondrocytes, adipocytes

## Abstract

The aging of our population is accompanied by an increased prevalence of chronic diseases. Among those, liver, joint and adipose tissue-related pathologies have a major socio-economic impact. They share common origins as they result from a dysregulation of the inflammatory and metabolic status. Plant-derived nutrients and especially polyphenols, exert a large range of beneficial effects in the prevention of chronic diseases but require clinically validated approaches for optimized care management. In this study, we designed an innovative clinical approach considering the metabolites produced by the digestive tract following the ingestion of an artichoke leaf extract. Human serum, enriched with metabolites deriving from the extract, was collected and incubated with human hepatocytes, human primary chondrocytes and adipocytes to determine the biological activity of the extract. Changes in cellular behavior demonstrated that the artichoke leaf extract protects hepatocytes from lipotoxic stress, prevents adipocytes differentiation and hyperplasia, and exerts chondroprotective properties in an inflammatory context. These data validate the beneficial health properties of an artichoke leaf extract at the clinical level and provide both insights and further evidence that plant-derived nutrients and especially polyphenols from artichoke may represent a relevant alternative for nutritional strategies addressing chronic disease issues.

## 1. Introduction

The aging of our population is accompanied by an increased prevalence of chronic diseases that mainly originates from metabolic and inflammatory disorders. Thus, along with cardio-vascular conditions, osteoarthritis, obesity and liver diseases represent major burdens for populations and health care systems. Liver diseases account for approximately 2 million deaths per year worldwide and about 50% of people over 65 years old suffer from osteoarthritis [1,2,3].

Chronic diseases result from metabolic, genetic, inflammatory or environmental causes and most of the time, these causes interconnect. Approximately 2 billion adults are obese or overweight, laying the path for adipocytes and hepatocytes dysfunctions and increased exposition to inflammatory mediators as well [1]. Pharmacological treatments are effective overall but have been pointed out for their side effects. In this context, preventive approaches are of growing interest. Data from the literature over the last three decades support the hypothesis that diet is a major environmental factor impacting potently and, lastingly, the biological functions of an organism. Thus, nutritional prevention has become a major field of scientific investigation and is now considered a powerful alternative to manage metabolic and inflammatory chronic conditions.

Among diet bioactives, polyphenols are widely acknowledged for their health benefits, especially for the prevention of oxidative stress. In this study, we questioned whether polyphenols from artichoke leaf extract (ALE) might also benefit against metabolic and inflammatory-related diseases, including liver, adipose and cartilage issues.

In the literature, artichoke leaf extract (Cynara scolymus) has already been described as an antioxidant [4], choleretic and a hepatoprotective agent [5], as well as a lipid metabolism regulator [6,7]. The most important benefit seems to be supported by its hepatoprotective properties [8]. In healthy hypercholesterolemic adults, ALE treatment (1280 mg daily for 12 weeks) decreases plasma total cholesterol by 4.2% [9]. In patients with diagnosed NAFLD, ALE treatment (600 mg daily for 9 weeks) reduces liver size, serum total cholesterol and triglyceride concentrations [10]. In a rat model of a high-fat diet, ALE limits hepatic disorders by reducing the fatty liver deposition in the hepatic lobule [11].

Nevertheless, clinical approaches represent less than 1% of the investigations and remain mostly descriptive. Thus, the literature concludes on the lack of information on the mode of action and on the need for further investigations at a cellular level in humans.

In vitro models have been used to decipher such mechanisms, but they have shown the limitations of their physiological relevance. Indeed, “in real life”, cells never “see” an extract or any native molecules. Cells rather “deal with” metabolites that originate from food processing, digestion, absorption and transformation through the digestive tract.

To improve the relevance of cellular mechanism data in humans, we designed an original ex vivo clinical approach based on human serum enrichment, combining human metabolism and cell biology. This methodology was validated by two clinical studies, and its robustness was attested by three recent publications from our lab [12,13,14]. This pioneering approach considers metabolism at the whole-body level and includes the transformations of the extract that occur during the digestive track to decipher whether and how metabolites from ALE may clinically benefit both lipid metabolism in liver and adipose tissues and to osteoarthritis management as well.

The clinical study was carried out in two stages. The first phase allowing the characterization of the metabolites present in human serum after consumption of the ALE and determines the time frame of their absorption peak. A second phase dedicated to collect both naïve and enriched sera, before ingestion and at the absorption peak, respectively. Then, we determined the influence of sera enriched in metabolites of interest resulting from the consumption of the ALE on the behavior of human hepatocyte, adipocyte and chondrocyte cultures.

Changes in cellular activity demonstrated that ALE protects hepatocytes from lipotoxic stress, prevents adipocytes differentiation and hyperplasia and exerts chondroprotective properties in an inflammatory context. These data validate the beneficial health properties of the ALE at the clinical level and provide a better understanding of the mechanisms involved in mediating the hepatoprotective, metabolic and osteo-articular effects carried by an artichoke extract.

## 2. Materials and Methods

### 2.1. Ethics Clinical Trial

The investigations were carried out following the rules of the Declaration of Helsinki of 1975 (https://www.wma.net/what-we-do/medical-ethics/declaration-of-helsinki/, accessed on 1 April 2021) revised in 2013. The human study was approved by the French Ethical Committee (Comité de Protection des Personnes (CPP AU1618/N°IDRCB: 2020-A00628-31/Clermont-Ferrand—Sud-Est VI; approved 9 October 2020). The volunteers were informed of the objectives of the present study and of the potential risks of ingestion of the extract, such as diarrhea and abdominal pain.

### 2.2. Human Study Design

A pool of 10 healthy men (age: 26.8 years old, +/−5.0; BMI: 23.9 kg/m^2^, +/−2.3; >50 kg; without drug treatment; and no distinction on ethnicity) volunteered for this study. They were tested for normal blood formulation and for renal (urea and creatinine) and liver function (aspartate aminotransferase (AST), alanine aminotransferase (ALT), gamma-glutamyltransferase (GGT). Blood samples of all participants were obtained and collected in serum-separating tubes for serum collection. Biological samples were prepared, aliquoted and stored at the Centre d’Investigation Clinique de Clermont-Ferrand—Inserm 1405, a specialized research department that guarantees the quality of samples and compliance with regulatory and ethical obligations (certification according to the French standard NF S 96900).

The first step of the study aimed at determining the polyphenol absorption peak. Three healthy volunteers who fasted for 12 h were given at once 8 g of an artichoke leaf extract (ALE). The dose was set according to validated preclinical and clinical data [5,11,14,15,16,17]. The ALE extract was given dissolved in 200 mL of water. Approximately 6 mL of venous blood was collected from the cubital vein before and every 20 min after the ingestion for a total period of 240 min. Serum was prepared from venous blood samples and stored at −80 °C until analysis. Polyphenol absorption profiles were evaluated by NMR spectroscopy and the total antioxidant capacity of the serum samples (Appendix A). Once the absorption peak was determined, volunteers were called back for the collection of the enriched serum fraction. Ten healthy volunteers who fasted for 12 h were given at once 8 g of an artichoke leaf extract (ALE) dissolved in 200 mL of water. Approximately 78 mL of venous blood was drawn from the cubital vein before the ingestion for the collection of a naïve serum. Then, at the maximum absorption peak, 100 mL of blood was drawn for enriched serum collection. Serum was stored at −80 °C until analysis.

### 2.3. Artichoke Leaf Extract

The artichoke leaves (Cynara scolymus) were collected in France (Coutures) and processed by Evear Extraction. The aqueous extract of artichoke leaves was obtained from fresh leaves. The resulting liquor is concentrated and then vacuum oven dried to produce the artichoke leaf extract powder used in this study. The final product was characterized for its total phenolic content using a Folin–Ciocalteu method. A total of 1 mL of Folin–Ciocalteu reagent was added to 5 mL of diluted extract (2 mg/50 mL), the volume was raised to 50 mL by adding sodium carbonate (15% *w*/*v*). The absorbance of the solutions was measured using a spectrophotometer at 715 nm. Pyrogallol was used as the standard curve. The average total phenolic content (%) expressed in Pyrogallol reaches 27.8% ± 1.7 (SD). To get insight into the phenolic content, the chromatographic analysis was carried out according to the European Pharmacopoeia (Ph. Eur. 10.0, 2389 (04/2018)). External standards were used to identify the different metabolites (Appendix A).

### 2.4. NMR Spectroscopy

*Reagents.* Sodium phosphate monobasic (NaH_2_PO_4_), Sodium phosphate dibasic (Na_2_HPO_4_), Sodium Chloride (NaCl), 3- (trimethylsilyl)propionic acid-2,2,3,3-d4 sodium salt (TSP), Caffeic acid, Quinic acid and Deuterated oxide (D_2_O) were purchased from Sigma-Aldrich (St. Louis, MO, United States). *Buffer for serum samples.* The buffer solution was prepared by dissolving 118.80 mg of anhydrous NaH_2_PO_4_, 569.30 mg of anhydrous Na_2_HPO_4_, 450.00 mg of NaCl and 50.00 mg of TSP in 50 mL of D_2_O to obtain a 100 mM phosphate buffer at pH 7.4 containing 0.9% NaCl and 0.1% TSP. *Serum Preparation.* All the 39 serum samples were stored at −80 °C and defrosted 20 min before analysis. A total of 400 µL of the defrosted sample was vortexed and mixed with 100 µL of phosphate buffer prepared above, vortexed again and transferred to a 5 mm NMR tube. *Compound addition to serum.* A total of 400 µL of a defrosted basal serum sample was vortexed and mixed with 100 µL of phosphate buffer prepared above, vortexed again and transferred to a 5 mm NMR tube. After NMR analysis, the sample was removed from the NMR tube. A total of 0.5 mg of quinic acid and 50 µL of phosphate buffer were added, the obtained sample was vortexed again and transferred to a 5 mm NMR tube. Following the same procedure, 0.5 mg of caffeic acid was added to the sample. *NMR Spectroscopy.* All NMR experiments were performed at 300 K on a Bruker Avance III 600 MHz spectrometer equipped with a cryogenically cooled probe CPTXI. Samples were locked and shimmed automatically using IconNMR software (Bruker Biospin, Wissembourg, France), and randomly analyzed. Samples were analyzed by proton nuclear magnetic resonance (^1^H NMR). The spectra were collected with offset frequency in the middle of the spectrum coinciding with the water resonance, which was suppressed by prior saturation using continuous irradiation during relaxation delay. Two complementary experiments were recorded: one-dimensional ^1^H Nuclear Overhauser Effect spectroscopY NOESY1D (NOESY1dgppr sequence) [18] and Carr–Purcell–Meiboom–Gill CPMG (CPMGpr sequence) [19]. NOESY1D and CPMG were acquired with 128 scans, a recycle delay of 5.0 s, a spectral width of 12,019.230 Hz and 64 k data points. NOESY1D was acquired with a mixing time of 100 ms. Both 90° duration pulse and offset of the transmitter frequency were optimized automatically before each experiment using IconNMR software (Bruker Biospin, Wissembourg, France). Spectra were manually phased, baseline corrected and referenced to TSP at 0.0 ppm. Topspin software V3.2.0 and 4.0.2 (Bruker Biospin, Wissembourg France) were used for NMR data acquisition, processing and analyses.

### 2.5. Human Cell Cultures

#### 2.5.1. Human Hepatocyte Cultures

The human hepatocyte cell line HepG2 was obtained from the European Collection of Authenticated Cell Cultures and purchased from Sigma-Aldrich (85011430). For maintenance, HepG2 cells were cultured in Dulbecco’s modified Eagle’s medium (Invitrogen, Carlsbad, CA, USA) containing 10% fetal bovine serum (Invitrogen) and 1% penicillin/streptomycin (Life Technologies, Villebon-Sur-Yvette, France). Cells were grown at 37 °C in an atmosphere of 5% CO_2_/95% air in a cell culture flask. To analyze the effects of the ALE serum metabolites on a palmitate-induced lipidic stress, cells were preincubated for 24 h in DMEM in the presence of 10% of human enriched serum according to the Clinic’n’Cell protocol (DIRV INRA 18-00058) prior to an additional 48 h treatment with palmitate (Sigma, St. Louis, MO, USA) at 250 µM.

#### 2.5.2. Human Primary Adipocyte Cultures

Human subcutaneous pre-adipocyte cells were purchased from Lonza Group Ltd. Cells were cultured using Preadipocyte Growth Medium-2 Bullet Kit ^TM^ (Lonza Group Ltd., Basel, Switzerland). The initiation and expansion of the cells were performed using Preadipocyte Growth Medium-2 (Preadipocyte Basal Medium-2 supplemented with 10% FCS, 2 mM L-Glutamine, 30 µg/mL genistein and 15 ng/mL ampicillin). To induce differentiation, cells were plated at 10,000 cells/cm^2^ in Preadipocyte Growth Medium-2. After 24 h, the culture medium was changed to adipocyte differentiation medium consisting of Preadipocyte Growth Medium-2 supplemented with insulin, dexamethasone, indomethacin and isobutyl-methylxanthine. Cells were allowed to differentiate for 8 days with no further media change. FCS was replaced by human enriched serum (DIRV INRA 18-00058) during the whole differentiation process in order to evaluate the effects of the ALE serum metabolites on adipocyte differentiation.

#### 2.5.3. Human Primary Chondrocyte Cultures

Human articular chondrocytes (HACs) were harvested from the tibial plateau and femoral condyles following knee replacement surgery and isolated as previously described [14]. Only intact cartilage areas were kept and processed for chondrocytes isolation. Briefly, cartilage was sliced, and chips were successively digested at 37 °C with 0.05% type IV-S hyaluronidase (750–3000 units/mg) (Sigma-Aldrich, Lyon, France) in Hank’s Balanced sodium Salt (HBSS) (Life Technologies, Villebon-Sur-Yvette, France) for 10 min and then with 0.2% trypsin (≥9000 BAEE units/mg) (Sigma-Aldrich, Lyon, France) for 15 min and with 0.2% type II collagenase (125 units/mg) (Sigma-Aldrich, Lyon, France) for 30 min. Cartilage chips were then digested overnight at 37 °C in 0.03% type II collagenase in the control medium (DMEM supplemented with 10% Fetal calf serum (FCS) (Pan-Biotech, Aidenbach, Germany) and 1% penicillin/streptomycin (P/S; Life Technologies, Villebon-Sur-Yvette, France). Cells were plated at passage 1 in F225 flasks at a density of 100,000 cells/cm² and maintained at 37 °C in a humidified atmosphere of 5% CO_2_ in the control medium (10% FCS, 1% P/S). At confluency, cells were subcultured for ex vivo experiments. After a 24 h incubation with ALE serum metabolites (DIRV INRA 18-00058), cells were further challenged with an additional 24 h treatment with human recombinant IL-1β (Millipore Corporation, Molsheim, France) at 1 ng/mL.

### 2.6. Cell Proliferation

The ex vivo cell proliferation was determined using an XTT-based method (Cell Proliferation Kit II, Sigma-Aldrich, Lyon, France) according to the supplier’s recommendations. Optical density was measured at 450 nm.

### 2.7. Preparation of Palmitate Solution

Palmitate (Sigma, St. Louis, MO, USA) stock solution was prepared by coupling palmitate to bovine serum albumin (BSA; Sigma). Palmitate was fully dissolved in pure ethanol at 70 °C for a concentration of 500 mmol/L. This palmitate stock solution was then added to a prewarmed BSA solution (10% *w*/*w*, 37 °C) to achieve a final palmitate concentration of 5 mmol/L. The solution was dissolved by incubating at 55 °C in a water bath for 15 min, twice. The final molar ratio of palmitate to BSA was 3.2:1. The control vehicle was prepared using a stock of 10% *w*/*w* BSA with an equivalent volume of ethanol added to match that contained in the final palmitate stock. The final concentration of ethanol was <0.05% by volume in all experiments.

### 2.8. Red Oil Staining

Oil Red O solution (0.5% in isopropanol) was purchased from Sigma, and staining was done following the supplier’s recommendations. Briefly, 3 parts of Oil Red O solution were mixed with 2 parts of water prior to the experiment to generate the working solution (0.2% in 60% isopropanol). Cells were washed twice with PBS before fixation with 4% paraformaldehyde (30 min at room temperature). The paraformaldehyde solution was then discarded, and cells were washed twice with water before being incubated with a 60% isopropanol solution for 5 min and with the working Oil Red O solution for 20 min. After 5 more washes with water, the cells were observed under the microscope. In order to quantify the staining, the fixed dye was redissolved in a fixed volume of 100% isopropanol, and the optical density was measured at 490 nm on an EL_X_808 IU spectrophotometer (BioTek Instruments, Colmar, France).

### 2.9. Cholesterol Levels

In HepG2 cells, the cholesterol levels were evaluated in both cell lysate and cell culture supernatant using the Cholesterol Quantification Kit from Sigma according to the manufacturer’s recommendations. The total cholesterol concentration was determined by a coupled enzyme assay, which results in a colorimetric (570 nm)/fluorometric (λ_ex_ = 535 nm/λ_em_ = 587 nm) product, proportional to the cholesterol present.

### 2.10. Triglycerides Levels

Triglyceride content was determined in both HepG2 and human subcutaneous pre-adipocyte cells using the Triglyceride Quantification Colorimetric/Fluorometric Kit from Sigma, according to the manufacturer’s protocols. TG is converted to free fatty acids and glycerol. The glycerol is then oxidized to generate a colorimetric (570 nm)/fluorometric (λ_ex_ = 535 nm/λ_em_ = 587 nm) product.

### 2.11. Real-Time RT-PCR

RNAm from either HepG2 cells or primary human adipocytes were isolated using TRIZOL according to the supplier’s recommendations. UCP-1, PPARγ and FABP4 mRNA expression levels in primary human adipocytes, as well as CYP7A1 and LDLR mRNA expression levels in HepG2 cells, were measured by RT-PCR (PowerUp SYBRgreen, Applied Biosystems). β-Actine was used as a housekeeping gene. Primers were designed as follows: UCP-1-F: GTC TGC CTT GAC TTT GAC AGT; UCP-1-R: AGG ACC AAC GGC TTT CTT C; PPARγ-F: GGA TTC AGC TGG TCG ATA TCA C; PPARγ-R: GTT TCA GAA ATG CCT TGC AGT; FABP4-F: ATC ACA TCC CCA TTC ACA CT; FABP4-R: ACT TGT CTC CAG TGA AAA CTT TG; CY7A1-F: GCT TTC ATT GCT TCT GGG TTC; CYP7A1-R: GAT GAT CTG GAG AAG GCC AAG; LDLR-F: AAA GTT GAT GCT GTT GAT GTT CT; LDLR-R: TGG CAG AGG AAA TGA GAA GAA G; ACTβ-F: ATT GGC AAT GAG CGG TTC; and ACTβ-R: GGA TGC CAC AGG ACT CCA.

### 2.12. NO, PGE2, and MMP-13 Quantification

Nitrate/Nitrite colorimetric assay and prostaglandin E2 Enzyme Immunoassay (EIA) kits were obtained from Cayman Chemical (Ann Arbor, MI, USA), and rabbit and human ELISA Kits for MMP-13 detection were purchased from Cloud-Clone Corp (Houston, TX, USA) and Abcam^®^ (Paris, France), respectively. The NO, PGE2 and MMP-13 level measurements were performed according to the manufacturer’s instructions. For human serum, measurements were performed in quadruplicates for each sample of the ten volunteers.

### 2.13. GAG Assay

A dimethylmethylene blue (DMB) assay was used to detect GAG production in cell lysates as previously described [20]. DMB solution was prepared at a final concentration of 46 mmol/L in a pH 3 adjusted buffer: 40 mmol/L NaCl, 40 mmol/L glycine. Sample concentrations were determined by mixing 50 µL of cell extract with 200 µL of DMB reagent. Following 30 min incubation, the absorbance was read at 595 nm on an EL_X_808 IU spectrophotometer (BioTek Instruments, Winooski, VT, USA). The GAG content was determined using a standard curve of chondroitin sulfate (Sigma). Results are expressed as µg of GAG per mg of total proteins determined by BCA assay (Sigma).

### 2.14. Cell Lysis

Cells were lysed using lysis buffer (50 mmol/L Tris pH 7.8, 150 mmol/L NaCl, 0.5% sodium deoxycholate, 1% NP40), and each fraction was stored at −80 °C until analysis.

### 2.15. Protein Quantification

Protein contents were determined by the BCA Protein Assay Kit (Millipore 71285-M). The BCA protein assay is based on a biuret reaction, which is the reduction of Cu^2+^ to Cu^+^ by proteins in an alkaline solution with concentration-dependent detection of the monovalent copper ions. Bicinchoninic acid is a chromogenic reagent that chelates the reduced copper, producing a purple complex with strong absorbance at 562 nm.

### 2.16. Statistics

Each experiment was performed at least in triplicate. Results are expressed as mean +/− SEM (standard error of the mean). Statistical analyses were carried out using ExcelStat Pro (Microsoft, Issy-les-Moulineaux, France). One-way ANOVA followed by Tukey’s test or T-test were performed. Groups with significant differences (*p* < 0.05) are indicated with different letters or * versus.

## 3. Results

### 3.1. Kinetic Profile of the ALE Absorption

To determine the time frame of the ALE absorption, three fasted volunteers were given the extract. Then blood was collected every 20 min for 4 h, and the resulted sera were analyzed by ^1^H NMR spectroscopy. Figure 1 shows representative ^1^H NMR spectra of basal human serum, as measured by 600 MHz ^1^H NMR spectroscopy. The spectra display a wide range of metabolites, such as phenolic compounds, sugars, amino acids and other metabolites.

Both global and pairwise comparisons of each volunteer ^1^H NMR spectra at all collection times were performed and revealed apparent metabolic changes in serum profiles between pre- and post-extract consumption (Figure 2A,B). In agreement with the high level of phenolic compounds in the ingested ALE, the overlay of ^1^H NMR spectra as a function of time showed that the highest variations were observed in the phenolic compounds’ region and for the 100 min samples (Figure 2A,B and Appendix A). This time was thus considered as the ALE absorption peak.

To gain further insights into the changes observed in the volunteer metabolic profiles, caffeic and quinic acids, which are the degradation products of the major polyphenol compound of the ALE (chlorogenic acid) [21], were added to a basal time serum sample. Caffeic and quinic acid addition provoked similar chemical shift variations to those observed for the 100 min samples (Figure 3A,B). These observations suggest that the ingestion of artichoke leaf extract has an effect on the global metabolome and that the changes observed after extract consumption are mainly due to polyphenols or their degradation products. Considering the aromatic composition of these compounds, the effect is more apparent in the phenolic compounds zone.

As a result, enriched serum was collected 100 min after the intake of artichoke leaf extract and used in clinical trials.

### 3.2. Influence of Human Serum Incubation on Cell Model Proliferation

To ensure the consistency of our ex vivo approach, we checked the influence of the different human sera on cell proliferation and viability in all our culture models, and we compared it to a regular fetal calf serum treatment by measuring XTT-based activity. As expected, cell growth stopped in serum-free cultures while cells proliferated in the presence of FCS 10% (hepatocytes +143% between 24 and 48 h; primary adipocytes +75% between 24 and 72 h; primary chondrocytes +29% between 24 and 72 h/(Figure 4A,C,E). Naïve or enriched human serum processed according to the Clinic’n’Cell methodology (DIRV#18-0058; see the Patents section) did not exert any cytotoxic effect on cell growth as compared to regular fetal calf serum and allowed cell proliferation similar to 10% FCS treatment (hepatocytes +142% between 24 and 48 h; primary adipocytes +212% between 24 and 72 h; primary chondrocytes +21% between 24 and 72 h (Figure 4B,D,F). These data validate further investigations.

### 3.3. Human Metabolites from ALE Prevent Human Hepatocytes from Lipotoxic Stress

Hepatocytes were exposed to a high concentration of palmitate (250µM) to induce lipotoxic stress. As expected and supported by the stronger red oil staining under palmitate treatment, hepatocytes accumulated intracellular fats (Figure 3). Lipid accumulation was massive with both types of serum, FCS and naïve human serum, thus validating the lipotoxic protocol (+675% and +246% for FCS and naïve human serum, respectively) (Figure 5A–D). In the presence of serum containing ALE metabolites, this accumulation was reduced by 34% compared to the lipotoxic control condition and limited to +130% compared to naïve serum condition meaning that the rise in lipid accumulation was almost divided by two (−47%) (Figure 5C–F).

Regarding the type of fats accumulated, we checked for intracellular TG and cholesterol contents. Consistently with red oil staining, the presence of palmitate increased intracellular TG similarly in both types of serum (FCS +278%; naïve human serum +289%). In the presence of serum containing ALE metabolites, TG accumulation tends to be reduced (−27%) compared to the lipotoxic control condition (Figure 6A,B), but data were not significant (*p* = 0.116).

The same trends were observed for cholesterol, with a palmitate-induced rise of intracellular cholesterol reaching +226% with FCS and +733% with naïve human serum. Compared to the lipotoxic control condition, the presence of serum containing ALE metabolites induced an abrogation of this rise by a magnitude close to the one observed for TG (−36%). In this case though, the observed effect was of statistical significance (Figure 6C,D).

Then, we checked the RNA expression level of two major actors involved in cholesterol metabolism and flux in hepatocytes, namely LDLR and CYP7A1, using real-time quantitative PCR. Both were reduced by the lipotoxic environment, but the presence of ALE metabolites countered the palmitate-induced down-regulation of both either way (LDLR, *p* = 0.012; CYP7A1, *p* = 0.060; Figure 7).

### 3.4. Human Serum Containing Metabolites from ALE Counter Adipocyte Differentiation

Primary human pre-adipocytes were committed to full differentiation using a commercially available differentiation kit (Lonza, Materials and Methods 2.6.2). As shown in Figure 8, when stimulated by the differentiation cocktail in the control FCS condition, red oil staining appears quite strong (+55%) when compared to FCS only.

In the absence of the differentiation cocktail, the presence of human serum enhanced, by itself, the red oil staining by +26% compared to the control FCS. In these conditions, it is worth noting that the presence of metabolites had no effect on this rise. Upon stimulation by the differentiation cocktail, staining increased by +32% compared to human serum alone. Interestingly, the enriched serum significantly inhibited this augmentation with −22% compared to naïve serum and limited this accumulation of fat to +8% compared to unstimulated control human serum condition.

Adipocytes mainly accumulate TG upon differentiation and hypertrophy; thus we checked for intracellular TG contents. Consistent with red oil staining, when primary human pre-adipocytes were committed towards maturation, TG accumulation was observed in both types of serum (FCS +830%, Figure 9A; naïve human serum +272%, Figure 9B). In the presence of serum containing ALE metabolites, TG accumulation was more than divided by two, as shown by a reduction of 54% compared to the differentiation condition (Figure 9B).

To elucidate further if and how ALE metabolites may limit adipocyte differentiation, we investigated the RNA expression level of white adipocyte markers, namely AP2 and PPARγ, using real-time quantitative PCR (Figure 10). As expected, these two markers were strongly increased in differentiated cells, and in both cases, this rise was very significantly reduced by the presence of ALE metabolites in the serum (AP2, −46%, *p* < 0.0001; PPARγ, −37%, *p* < 0.01).

It has been shown that a reduced differentiation toward white adipocyte lineage may be due to a switch toward brown adipocyte lineage. To test this hypothesis, the RNA expression of UCP1, a well-known marker of brown adipocytes, was evaluated. The presence of ALE metabolites in the serum had no effect on UCP1 RNA expression (Figure 8).

### 3.5. Human Serum Containing Metabolites from ALE Protects from Chondrocytes from Inflammatory Stress

As a classic arthrosis model, human primary chondrocytes were exposed to the pro-inflammatory cytokine IL1-β (1 ng/mL). As expected, there was an increased secretion of pro-inflammatory mediators (PGE2 +15,200% and NO +71%) by chondrocytes in response to the IL1-β stimulation in the presence of 10% FBS (Figure 11A,B). The rise was similar (PGE2 +15,600%, NO +56%) when the cells were cultured with 10% naive human serum (Figure 11C,D). Whereas the serum enriched with ALE metabolites had no effect on NO production (Figure 11D), it caused a significant reduction (−34%) of the PGE2 production in response to IL1-β (Figure 11C).

To understand where those effects on inflammatory mediators could impact the chondrocytes’ primary function, we evaluated the effect of the enriched serum on the production of molecular actors involved in either cartilage catabolism (MMP13) or cartilage matrix production (Glyco-Amino Glycans, GAGs). When the cells were cultured with 10% FBS, IL1-β induced a significant increase in MMP13 production (+1550%/Figure 12A), as well as a significant reduction of GAGs levels (−64%/Figure 12B). The same observations could be made in the presence of naive serum (MMP13, +765%; GAGs, −43%/Figure 12C,D). Enrichment of the human serum with ALE metabolites resulted, in IL1-β-stimulated cells, in a significant reduction of MMP13 production (−46%/Figure 12B), as well as in a rescued GAGs production similar to control cells (Figure 12D).

## 4. Discussion

Several clinical studies have demonstrated the health benefits of artichoke leaf extract on liver disease, obesity and inflammation-related dysfunctions. Most of the clinical data remain descriptive regarding the mechanism of action that is mainly linked to the antioxidant capabilities of the extract. Indeed, ingesting artichoke leaf extract results in higher plasma total antioxidant capacity (Appendix A) [22]. In our study, we found similar results, and in this light, we investigated whether and how human metabolites from ALE could support health effects targeting three different tissues that may share common metabolic and inflammatory stresses: liver, fat and cartilage.

At the cellular level, we found that: (1) ALE metabolites exert hepatoprotective properties in human hepatocytes by counteracting a fatty acid-induced lipotoxic stress that is commonly observed in hypercholesterolemia [9] and steatohepatitis [10]; (2) ALE metabolites limit adipogenic differentiation and hypertrophy, a cellular feature of obesity and metabolic syndrome [23]; (3) finally, ALE metabolites protect chondrocyte from an IL-1β stimulation that mimics osteo-arthritis inflammatory environment [14].

Our clinical ex vivo approach is integrative and considers the systemic aspect of digestion, in particular the hepato-intestinal barrier. It allows the testing of all the metabolites found in serum after ALE ingestion. Consequently, we keep intact the potential nutritional synergies that can occur at the whole-body level, but it remains difficult with our results to associate an isolated metabolite to a biological effect. Furthermore, we cannot exclude any contribution of endogenous anti-inflammatory or antioxidant mediators. Nevertheless, we did not observe the appearance of glutathione characteristic signals after ALE ingestion, indicating that its concentration is not significantly increased (Appendix A). Furthermore, when degradation products of chlorogenic acid were added to a naive serum sample, they led to similar chemical shift variations to those observed for the 100 min serum samples. The parallel between NMR signals strongly suggests that the apparent metabolic changes in the serum profile after ingestion are due to the presence of the ALE polyphenol metabolites or their degradation products. Thus, in the following parts, we compared the benefits that we observed at the cellular level with the literature data regarding the described effects of the different constituents of the ALE.

Chlorogenic acid may be isolated from ALE. It protects hepatocytes from FFA-induced lipotoxicity through activation of SIRT1 [24]. Luteolin attenuates hepatic lipotoxicity and reduces chronic low-grade inflammation by modulating the TLR signaling pathway [25]. High-dose aqueous extracts from artichoke leaves were found to inhibit cholesterol biosynthesis from ^14^C-acetate in primary cultured rat hepatocytes [26]. Flavonoids from artichoke, including luteolin, stimulate biliary secretion and therefore cholesterol elimination [27,28]. Luteolin improves hypercholesterolemia and glucose intolerance in diet-induced obese mice and stimulates cholesterol efflux in HepG2 hepatocytes through the LXRα-dependent pathway [29]. A recent review concluded that polyphenols might induce CYP7A1 expression predominantly by the LXRα pathway in rodents cells while this may occur through FXR, NF-KB and ERK signaling in human cells [30].

Although consistent with our results, these in vitro experiments were conducted with extracts or purified molecules, pushing the results away from a physiological point of view. In fact, tissues and cells never interact with such extracts or native molecules. Here, we demonstrated that human metabolites from ALE protect human hepatocytes from palmitate-induced lipotoxic stress. In this well-acknowledged cellular model, mimicking steatohepatitis, ALE metabolites were found to limit the intra-cellular fat accumulation and support bile acid biosynthesis and cholesterol elimination via a preserved expression of both LDLR and CYP7A1.

In a very recent paper published in 2021 from Liao et al., the authors demonstrated, in a mice model of steatohepatitis and liver damages, that ALE ameliorates both hepatic oxidative stress and lipid metabolism disorder [31]. Together with hepatocytes, adipocytes are key regulators of metabolic and inflammatory disorders. Then, deciphering how ALE may influence primary human adipocyte behavior and lipid metabolism contribute to the understanding of its health benefits. In high-fat diet rodent models, ALE limits body and organ weight gain by restraining serum total cholesterol and triglycerides and cell hypertrophy [32]. Chlorogenic acid (CGA) may account for this effect, as it can reduce body weight and fat deposition in a monosodium glutamate (MSG)-induced obesity mouse model [33]. Moreover, CGA has been described as an inhibitor of adipocyte differentiation with decreased lipids and triacylglycerol accumulation in 3T3-L1 cells and downregulation of the expression of peroxisome proliferator-activated receptor-gamma (PPARγ) [34]. Here, consistently with the literature data, ALE metabolites potently prevented adipocytes from lipid and, more specifically, TG intra-cellular accumulation. This inhibition of adipocyte hypertrophy was associated with the down-regulation of PPARγ, a transcription factor promoting white adipocyte differentiation, and AP2 (FABP4), the main marker of differentiated white adipocytes.

Then we questioned whether ALE metabolites were able to switch the differentiation process and brown our primary human adipocytes. Indeed, luteolin-enriched ALE alleviates the metabolic syndrome in mice with high-fat diet-induced obesity by decreasing lipogenesis while increasing fatty acid oxidation [16]. Chlorogenic acid enhances the thermogenesis and proton leak of C_3_ H_10_ T_1/2_ cells [35]. Mechanistically, dietary luteolin activates browning and thermogenesis in primary mouse adipocytes through an AMPK/PGC1alpha pathway-mediated mechanism [36]. In the same trend, the combination of chlorogenic and caffeic acids induces lipolysis, upregulates AMPK and browning gene expression (including UCP1) while downregulating peroxisome proliferator-activated receptor γ (PPARγ) at both transcriptional and protein levels [37]. In this ex vivo clinical study, we did not find any up-regulation of UCP-1 expression despite the down-regulation of PPARγ and AP2 (FABP4). Thus, one may speculate that the aforementioned browning may be observed with purified molecules that did not undergo the metabolic transformation operated by enterocytes or hepatocytes on polyphenols, including sulfation, glucuronidation and methylation [38].

Interestingly, the ALE-related prevention of hepatic oxidative stress and lipid metabolism disorder observed in a mouse model of steatohepatitis was associated with a lower IL-1β expression that accounts for osteoarthritis onset [14,31]. Indeed, the dysmetabolic and inflammatory context responsible for liver and adipose tissues conditions is also the driving force for osteoarthritis establishment. Thus, ALE benefits that apply to hepatocytes and adipocytes may be extended to chondrocytes. Cynaropicrin, present in ALE, is a dual regulator for both degradation and synthesis factors in cartilage metabolism. ALE and cynaropicrin both suppress IL-1β-induced NF-κB signaling and inhibit the production of MMP13 in the human chondrosarcoma cell line OUMS-27 [17]. While these data are parallel with ours, this team observed that cynaropicrin decreased the synthesis of aggrecan. In contrast, in ours, metabolites from ALE prevented the loss of glycoaminoglycans (GAG) production induced by IL-1β and even stimulated their production in the absence of pro-inflammatory stimuli (data not shown). Once again, this seeming discrepancy may rely on the differential activity between a native nutrient and its metabolites depending on the biological target. According to the chondroprotective effects of ALE metabolites with regard to the IL-1β-driven stress, luteolin attenuates osteoarthritis progression and collagen II degradation in a rat model of monoiodoacetate-induced osteoarthritis [39]. In C57BL/6N mice fed a high-fat diet, luteolin inhibits IL-1β-induced inflammation [16]. In vitro, it effectively decreases NO, TNF-α and IL-6 production in chondrocytes from guinea pigs and down-regulates the expressions of JNK and p38MAPK [40]. Other than luteolin, chlorogenic acid, another main constituent of ALE, prevents inflammatory responses, including NO and PGE2, in the same model we used for osteoarthritis using IL-1beta-stimulated human chondrocytes [41,42]. Consistently, in 2021, Matsuda et al. found that sesquiterpene lactones from artichoke leaves, including cynaropicrin, inhibit NO production and the induction of iNOS via JAK-STAT and NF-kB in osteoclast precursors RAW264.7 cells [43]. Furthermore, in an alginate scaffold of chondrocytes, chlorogenic acid can improve the repair of damaged articular cartilage [44].

In line with the literature data, we bring further biological and clinical evidence of the health benefit of ALE on inflammation-related chronic diseases providing insights on the role of human metabolites rather than nutrients or native molecules.

Other than polyphenols, ALE may contain very-long-chain inulin that is mainly extracted from globe artichoke (Cynara scolymus). Given at the dose of 10 g daily for 3 weeks, these very-long-chain inulin exert a bifidogenic effect on the human intestinal microbiota that could contribute to the health benefits of ALE [45]. However, our ex vivo clinical approach relies on acute exposure and does not consider chronic exposure. Metabolites from ALE obtained with our approach would unlikely involve such bifidogenic effects.

To keep this acute exposure safe and physiologically sound for this ex vivo clinical protocol, the dose was set according to the literature data regarding clinical and preclinical studies. In vivo studies have helped define avenues of investigation for determining the health benefits and the effective dose scales of ALE. In a mouse model of cirrhosis, daily gavage for 10 days with ALE at a dose of 1.6 g/kg body weight limits hepatic tissue degeneration [5]. In diabetic rats, the daily administration of ALE at 400 mg/kg body weight for 8 weeks, exerts anti-hyperglycemic, antioxidant and lipid-lowering effects [11,15]. These doses of 400 mg/kg/day in rats and 1.6 g/kg/day in mice for 10 to 56 days correspond to 8 to 15 g/day in humans according to the metabolic weight conversion table. In humans, the average dose for a chronic exposure is around 1.8 g, ranging from 250 to 3200 mg for 5 to 12 weeks (250 [46]; 1200 [9,22]; 600 [10]; 1800 [23,47]; 3200 mg [48]).

In this protocol, we used 8 g of ALE in a single dose format. That is only 2.5 times higher than the daily dose used for chronic exposure in Huber’s study (3.2 g for 12 weeks) [48]. The ALE contains approximately 28% of polyphenols. Thus, each volunteer received 2.2 g of polyphenols. This quantity is lower than a previous clinical study we published in 2019 with 3 g of polyphenols from olive and grape [14]. Furthermore, the recommendation for the polyphenol daily intake is around 1 g [21,49]. Therefore, even at a higher dose than used in this protocol is nutritional.

Other than the dose, the physiological relevance of such an ex vivo clinical approach is high as it allows the testing of all the metabolites at the same time and preserve the potential synergistic effects between nutrients from ALE. For instance, in a recent in vivo study, the anti-adiposity and anti-dyslipidemic effects of ALE were more pronounced than luteolin alone in mice fed a high-fat diet [16]. In parallel, as each volunteer is their own control, variation between individuals is limited. Therefore, this approach allows the study to reach a relevant statistical power with a small sample of human volunteers, as demonstrated by our previous publications [12,13,14]. This clinical ex vivo methodology was developed and protected by the French National Institute for Agronomic and Food Research and the Clermont-Auvergne University for a more rapid, physiological and ethical research in nutrition.

## 5. Conclusions

Finally, from a global perspective, our results correlate not only with the literature data on the positive role of ALE, but using a pioneering clinical ex vivo approach considering the digestive processes of nutrients, we give clues on the role of human metabolites from ALE and provide further biological and clinical evidence of the health benefit of ALE on inflammation-related chronic diseases including liver, adipose tissue and cartilage conditions.

## 6. Patents

The human ex vivo methodology used in this study has been registered as a written invention disclosure by the French National Institute for Agronomic Research (INRA) (DIRV#18-0058). Clinic’n’Cell^®^ has been registered as a mark [12,13,14].

## Figures and Tables

**Figure 1 nutrients-13-02653-f001:**
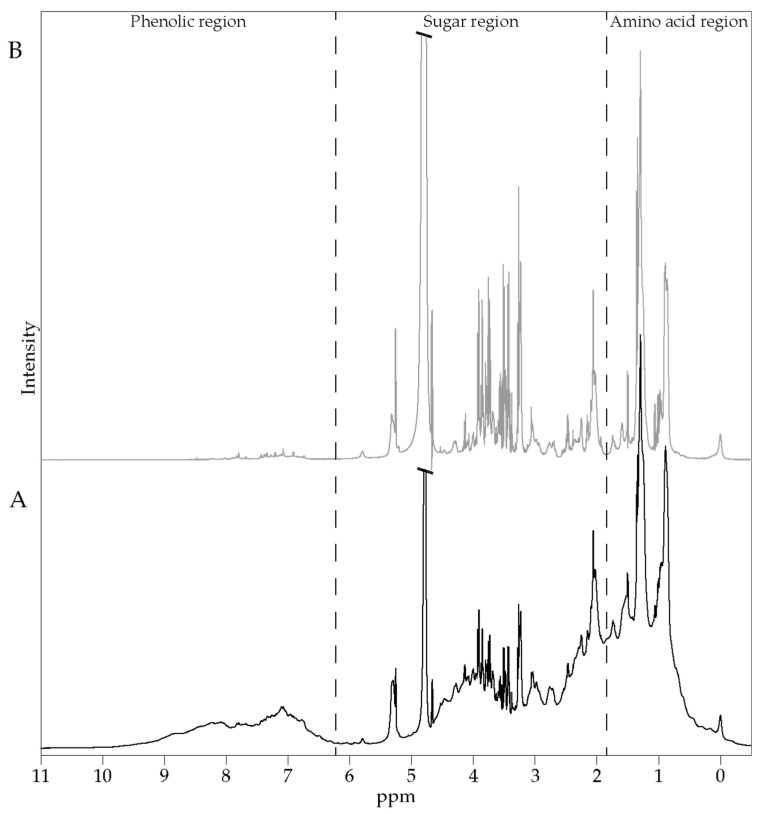
Typical 600 MHz ^1^H NMR spectra of a basal serum; 1D ^1^H NOESY (**A**) and 1D ^1^H CMPG (**B**) spectra. Regions containing characteristic signals of phenolic compounds, sugars and amino acids are indicated.

**Figure 2 nutrients-13-02653-f002:**
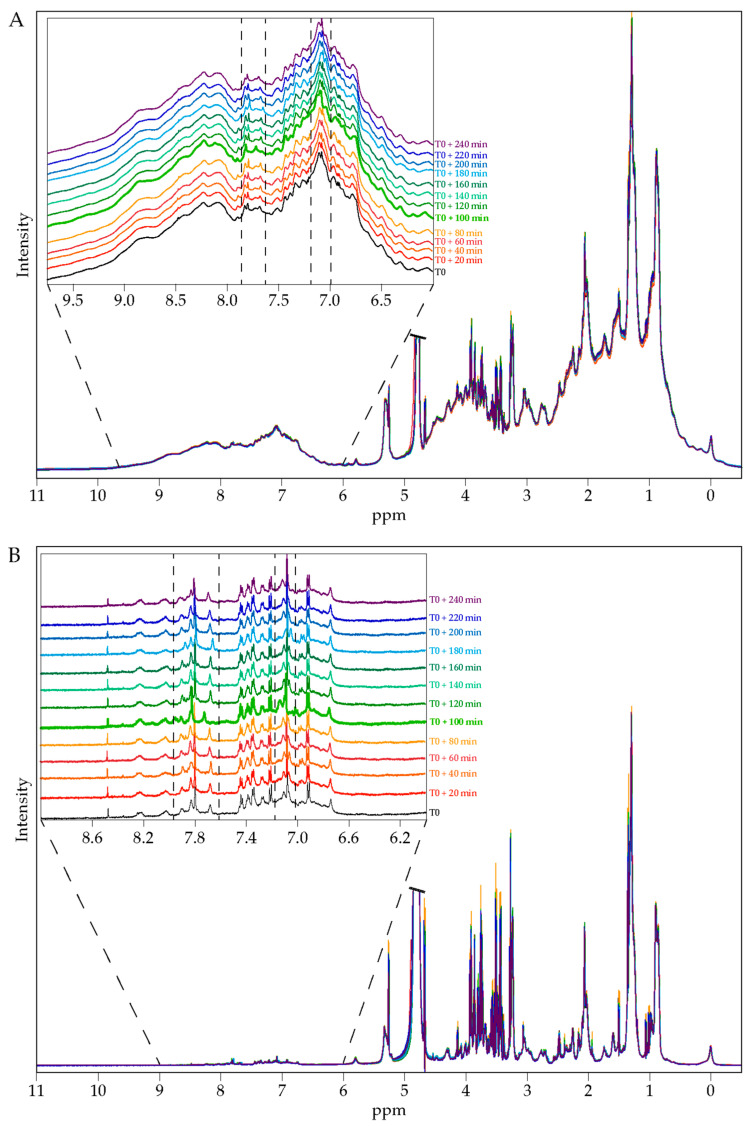
Overlay of ^1^H NMR spectra of serum of one volunteer before and after artichoke leaf extract consumption; 1D ^1^H NOESY (**A**) and 1D ^1^H CMPG (**B**). Inserts in each panel show an enlargement of the phenolic region.

**Figure 3 nutrients-13-02653-f003:**
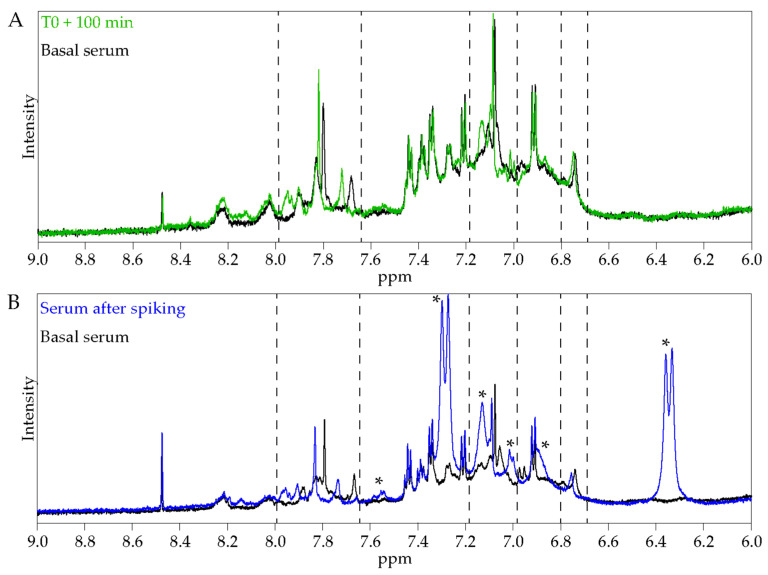
Expansion of the phenolic region of ^1^H NMR spectra of serum. (**A**) Expansion of the phenolic region of ^1^H NMR spectra of serum collected at T0 (black) and T0+100 min (green). (**B**) Expansion of the phenolic region of ^1^H NMR spectra of a basal serum (black) and after spiking with caffeic and quinic acids (blue). *: caffeic acid signals.

**Figure 4 nutrients-13-02653-f004:**
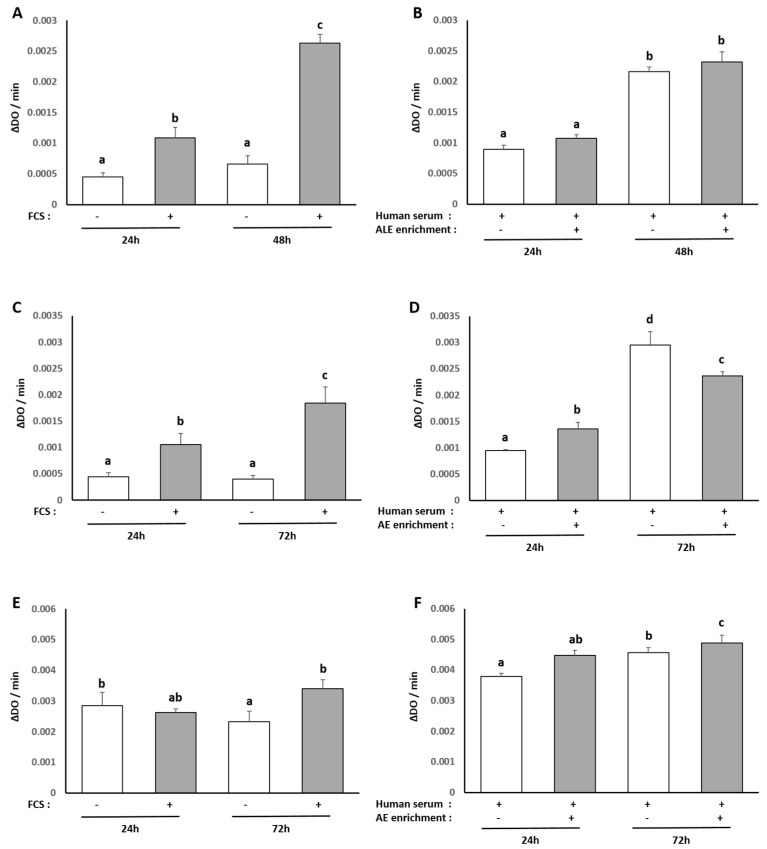
Effect of fetal calf serum and human serum enriched with ALE metabolites on the proliferation of human hepatocytes HepG2 cells, primary human subcutaneous pre-adipocytes and primary human articular chondrocytes: XTT activity. Human primary chondrocytes were incubated with serum from calf (**A**,**C**,**E**) or human origin (**B**,**D**,**F**) for 24 and 48 h. Primary human subcutaneous pre-adipocytes and HepG2 cells were incubated with serum from calf (**C**,**E**) or human (**D**,**F**) origin for 24 and 72 h. In each case, proliferation was measured using the XTT-based assay. Cells proliferate without any negative impact of human serum or metabolite enrichment (−absence; + presence). Groups with significant differences (*p* < 0.05) are indicated with different letters (a, b, c and d).

**Figure 5 nutrients-13-02653-f005:**
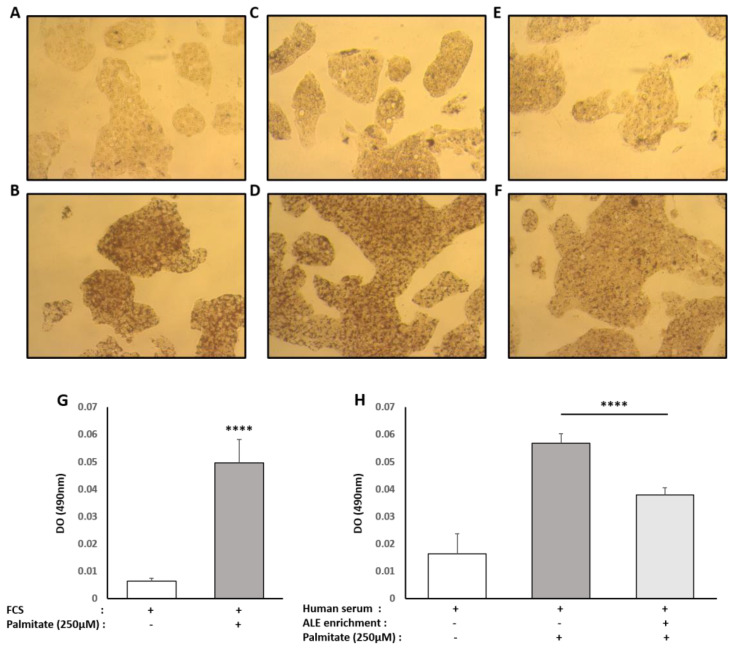
Effect of human serum enriched with ALE metabolites on total intracellular lipid content in HepG2 cells. HepG2 cells were preincubated with fetal calf serum (**A**,**B**), human naive serum (**C**,**D**) or ALE enriched human serum (**E**,**F**) for 24 h and stimulated with 250µM palmitate (**B**,**D**,**F**) or vehicle (**A**,**C**,**E**) for an additional 48 h. Intracellular lipid content was analyzed using an oil red coloration. After microscopical observation, the fixed dye was redissolved in 100% isopropanol, and the optical density was measured at 490 nm in order to quantify lipidic content (**G**,**H**). Palmitate increases intracellular lipid content independently of the origin of the serum (calf or human). The human serum enriched with ALE metabolites significantly limited the palmitate-induced steatosis. **** (*p* < 0.0001).

**Figure 6 nutrients-13-02653-f006:**
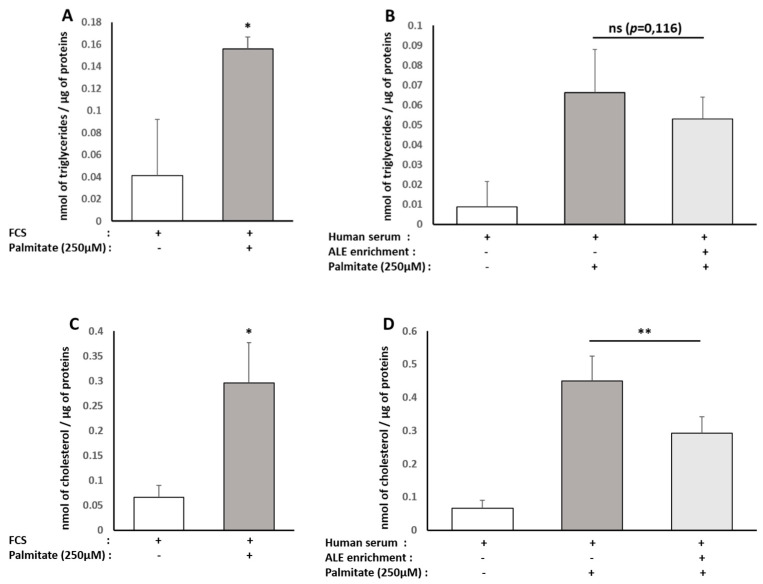
Effect of human serum enriched with ALE metabolites on cholesterol and triglyceride levels in HepG2 cells. HepG2 cells were preincubated with fetal calf serum for 24 h and stimulated with 250 µM palmitate or vehicle for an additional 48 h (**A**,**C**). To analyze the effects of human serum enriched with metabolites, cells were preincubated for 24 h in DMEM in the presence of 10% of human enriched serum according to the Clinic’n’Cell protocol (DIRV INRA 18-00058) prior to an additional 48 h treatment palmitate (Sigma, St. Louis, MO, United States) at 250 µM. Intracellular triglycerides (**A**,**B**) and total cholesterol were measured (**C**,**D**). Palmitate stimulated HepG2 triglyceride and cholesterol content independently of the origin of the serum (calf or human). The human serum enriched with ALE metabolites significantly limited the intracellular cholesterol levels. * (*p* < 0.05); ** (*p* < 0.01); *ns*: no significant difference.

**Figure 7 nutrients-13-02653-f007:**
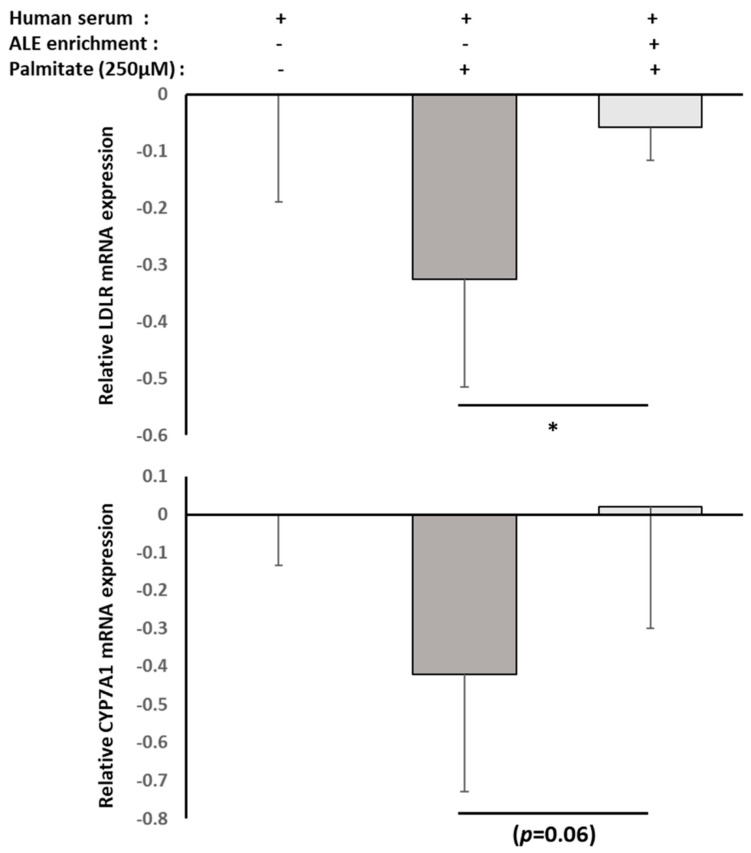
Effect of human serum enriched with ALE metabolites on LDLR and CYP7A1 mRNA expression levels in HepG2 cells. HepG2 cells were preincubated with human serum enriched with ALE metabolites for 24 h and stimulated with palmitate (250 µM) for an additional 48 h (− absence; + presence). LDLR and CYP7A1 mRNA expression levels in HepG2 cells were measured by RT-PCR (PowerUp SYBRgreen, Applied Biosystems). β-Actine was used as a housekeeping gene. Primers were designed as follows: LDLR-F: AAA GTT GAT GCT GTT GAT GTT CT; LDLR-R: TGG CAG AGG AAA TGA GAA GAA G; CYP7A1-F: GCT TTC ATT GCT TCT GGG TTC; CYP7A1-R: GAT GAT CTG GAG AAG GCC AAG; ACTB-F: ATT GGC AAT GAG CGG TTC; and ACTB-R: GGA TGC CAC AGG ACT CCA. Palmitate-induced inhibition of both LDLR and CYP7A1 mRNA expressions was reduced by the presence of ALE metabolites. Significance was reached for LDLR with * (*p* < 0.05). *p*-value for CYP7A1 is *p* = 0.06.

**Figure 8 nutrients-13-02653-f008:**
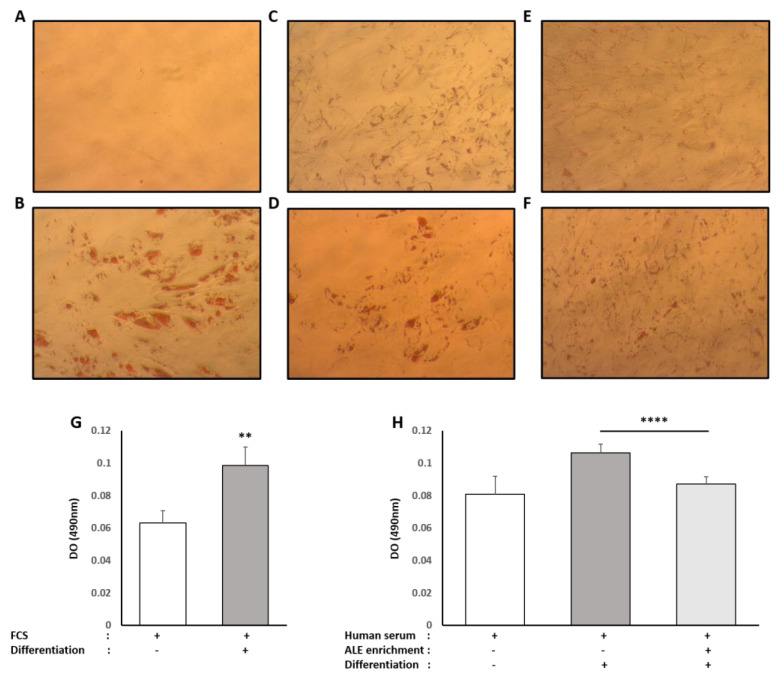
Effect of human serum enriched with ALE metabolites on total intracellular lipid content in primary human subcutaneous pre-adipocytes: Primary human subcutaneous pre-adipocytes were differentiated for 8 days in an adipocyte differentiation medium consisting of Preadipocyte Growth Medium-2, supplemented with insulin, dexamethasone, indomethacin and isobutyl-methylxanthine. To analyze the effects of human serum enriched with metabolites, cells were either differentiated in the presence of fetal calf serum (**A**,**B**), human naive serum (**C**,**D**) or ALE enriched human serum (**E**,**F**). Intracellular lipid content was analyzed using a red oil coloration. After microscopical observation, the fixed dye was redissolved in 100% isopropanol, and optical density was measured at 490 nm in order to quantify lipidic content (**G**,**H**). Differentiation increases intracellular lipid content independently of the origin of the serum (calf or human). The human serum enriched with ALE metabolites significantly limited the intracellular lipid accumulation in differentiated cells. ** (*p* < 0.01); **** (*p* < 0.0001).

**Figure 9 nutrients-13-02653-f009:**
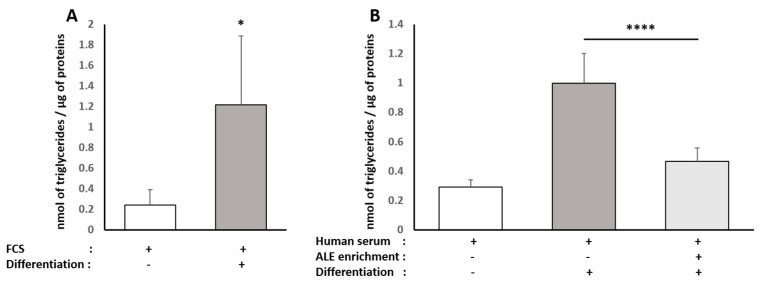
Effect of human serum enriched with ALE metabolites on triglyceride levels in primary human subcutaneous pre-adipocytes: Primary human subcutaneous pre-adipocytes were differentiated for 8 days in an adipocyte differentiation medium supplemented with either 10% fetal calf serum (**A**) or 10% human serum (**B**). Intracellular triglycerides were measured. Differentiation induces an increase in intracellular triglyceride content independently of the origin of the serum (calf or human). The human serum enriched with ALE metabolites significantly limited the intracellular triglyceride accumulation in differentiated adipocytes * (*p* < 0.05); **** (*p* < 0.0001).

**Figure 10 nutrients-13-02653-f010:**
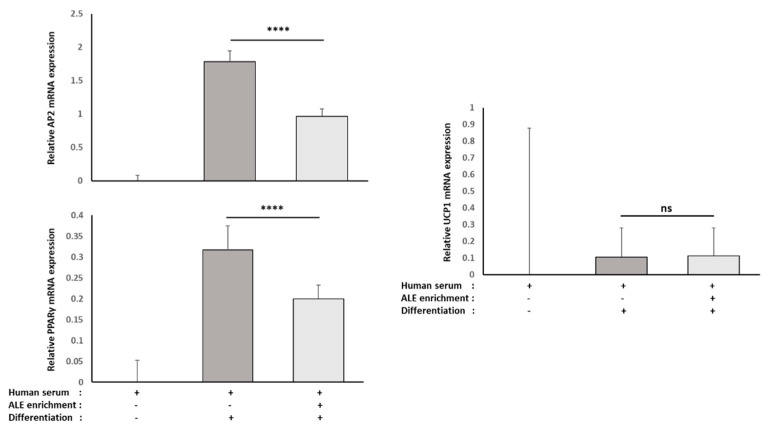
Effect of human serum enriched with ALE metabolites on AP2, PPARγ and UCP1 mRNA expression levels in primary human subcutaneous pre-adipocytes: Primary human subcutaneous pre-adipocytes were differentiated for 8 days in the presence of human serum enriched with ALE metabolites. AP2, PPARγ and UCP1 mRNA expression levels were measured by RT-PCR (PowerUp SYBRgreen, Applied Biosystems). β-Actine was used as a housekeeping gene. Primers were designed as follows: AP2-F: ATC ACA TCC CCA TTC ACA CT; AP2-R: ACT TGT CTC CAG TGA AAA CTT TG; PPARγ -F: GGA TTC AGC TGG TCG ATA TCA C; PPARγ -R: GTT TCA GAA ATG CCT TGC AGT; UCP1-F: GTC TGC CTT GAC TTT GAC AGT; UCP1-R: AGG ACC AAC GGC TTT CTT C; ACTB-F: ATT GGC AAT GAG CGG TTC; and ACTB-R: GGA TGC CAC AGG ACT CCA. Both AP2 and PPARγ mRNA expression were significantly reduced by the presence of ALE metabolites in differentiated cells. Based on UCP1 mRNA expression, the reduced white adipocyte differentiation did not involve a switch toward brown adipocyte differentiation. **** (*p* < 0.0001); *ns*: no significant difference.

**Figure 11 nutrients-13-02653-f011:**
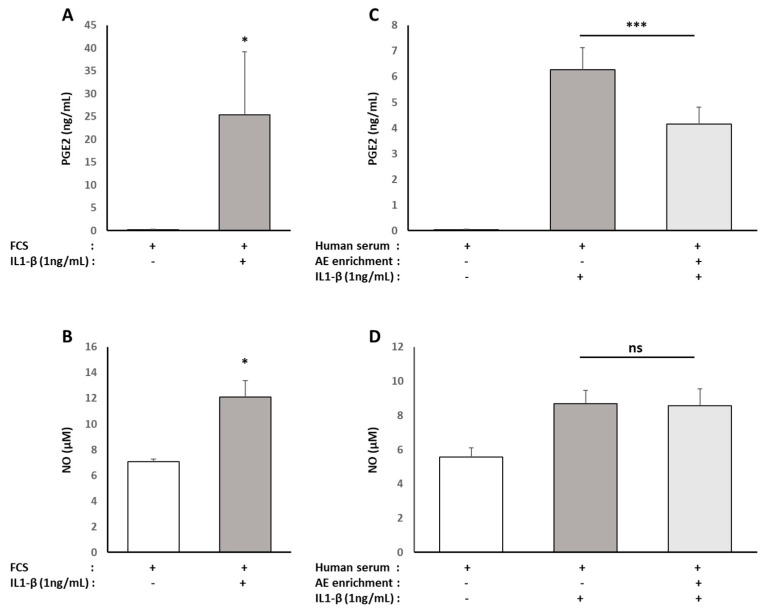
Effect of human serum enriched with ALE metabolites on inflammatory mediator production of NO and PGE2 in primary human articular chondrocytes: Human articular chondrocytes (HACs) were harvested from the tibial plateau and femoral condyles following knee replacement surgery and isolated. HACs were preincubated with fetal calf serum for 24 h and stimulated with IL-1 β (1 ng/mL) for an additional 24 h (**A**,**B**). To analyze the effects of human serum enriched with metabolites, cells were preincubated for 24 h in DMEM in the presence of 10% of human enriched serum according to the Clinic’n’Cell protocol (DIRV INRA 18-00058) prior to an additional 24 h treatment with human recombinant IL-1β (Millipore Corporation, Molsheim, France) at 1 ng/mL (**C**,**D**). NO (**B**,**D**) and PGE2 (**A**,**C**) releases in culture media were measured. IL-1 β stimulated NO and PGE2 releases independently of the origin of the serum (calf or human). The human serum enriched with OPCO metabolites significantly limited PGE2 release but had no observable effect on NO release. * (*p* < 0.05); *** (*p* < 0.001); *ns*: no significant difference.

**Figure 12 nutrients-13-02653-f012:**
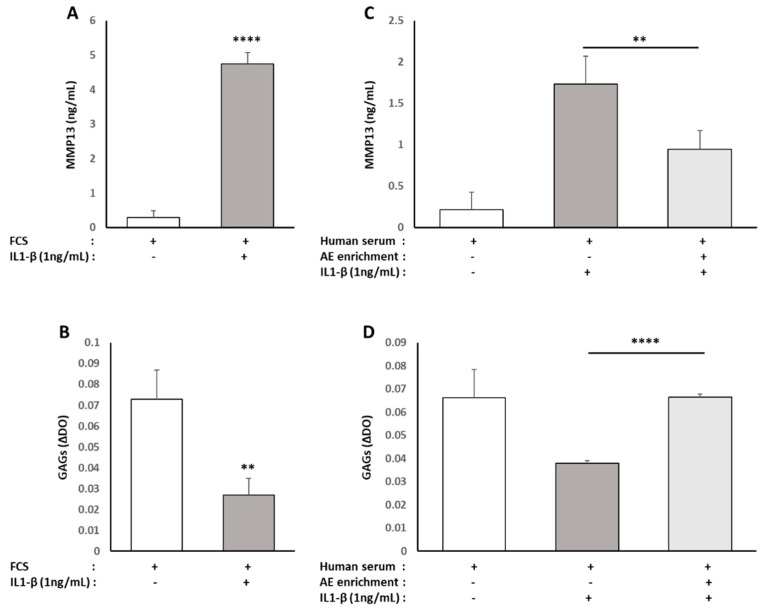
Effect of human serum enriched with ALE metabolites on catabolic actors (MMP13) and cartilage matrix production (Glyco-Amino Glycans/GAGs) in primary human articular chondrocytes: Human articular chondrocytes (HACs) were harvested from the tibial plateau and femoral condyles following knee replacement surgery and isolated. HACs were preincubated with fetal calf serum for 24 h and stimulated with IL-1 β (1 ng/mL) for an additional 24 h (**A**,**B**). To analyze the effects of human serum enriched with metabolites, cells were preincubated for 24 h in DMEM in the presence of 10% of human enriched serum according to the Clinic’n’Cell protocol (DIRV INRA 18-00058) prior to an additional 24 h treatment with human recombinant IL-1β (Millipore Corporation, Molsheim, France) at 1 ng/mL (**C**,**D**). MMP13 release in culture media (**A**,**C**) and GAG cellular content (**B**,**D**) were measured. IL-1 β stimulated MMP13 release and reduced GAGs cellular content independently of the origin of the serum (calf or human). The human serum enriched with OPCO metabolites significantly reduced MMP13 release and limited the loss of GAGs production. ** (*p* < 0.01); **** (*p* < 0.0001).

## Data Availability

The data presented in this study are available on request from the corresponding author. The data are not publicly available due to ethical restrictions.

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
