# Peer review of "Metabolic and Anti-Inflammatory Protective Properties of Human Enriched Serum Following Artichoke Leaf Extract Absorption: Results from an Innovative Ex Vivo Clinical Trial"

_nutrients, 2021, doi:10.3390/nu13082653_

Round 1

Reviewer 1 Report

This paper is very interesting to readers of this journal and to scientists in other disciplines, as well as its applicability. The article is very interesting and devoted to very important aspects related to the validation of the beneficial health properties of the Artichoke Leaf Extract at the clinical level and provide a better understanding of the mechanisms involved in mediating the hepatoprotective, metabolic and osteo-articular effects carried by an artichoke extract. However, I have several suggestions before publication of this manuscript

The article has the appropriate size, but it is necessary to better organize the text.

Abstract

The abstract appropriately cover the contents of the article. Indeed, I have several suggestions on the text of the paper, not on abstract.

Keywords

The key words are very difficult to understand. I think it should be simpler and easier to search.

Introduction

Introduction should be implemented emphasizing that the aging of our population is accompanied by an increased prevalence of chronic diseases with metabolic, genetic, inflammatory or environmental causes

Methods

The material and methods section are not very clear, so it could be structured differently. Sometimes they are a little redundant.

Results and Discussion

I suggest to emphasize the results and discussion of the analysis in the light of the results obtained in the established scientific literature.

In general, check the format of the text.

Author Response

This paper is very interesting to readers of this journal and to scientists in other disciplines, as well as its applicability. The article is very interesting and devoted to very important aspects related to the validation of the beneficial health properties of the Artichoke Leaf Extract at the clinical level and provide a better understanding of the mechanisms involved in mediating the hepatoprotective, metabolic and osteo-articular effects carried by an artichoke extract.

We thank you for such comments and compliments regarding our work, its interest, its design and its applicability.

Keywords: The key words are very difficult to understand. I think it should be simpler and easier to search.

According to your comment we changed keywords to: clinical trial; metabolic disorders; inflammation; artichoke; polyphenols; metabolites; NMR; human cells; hepatocytes; chondrocytes; adipocytes

Introduction: Introduction should be implemented emphasizing that the aging of our population is accompanied by an increased prevalence of chronic diseases with metabolic, genetic, inflammatory or environmental causes

We agree that key facts may contribute to better describe the societal context and the rationale of our study. As a consequence, the beginning of the introduction was implemented with epidemiologic data supported by 3 new very recent references.

Line 43: “The aging of our population is accompanied by an increased prevalence of chronic diseases that mainly originates from metabolic and inflammatory disorders. Thus, along with cardio-vascular conditions, osteoarthritis, obesity and liver diseases represent major burdens for populations and health care systems. Liver diseases account for approximately 2 million deaths per year worldwide and about 50% of people over 65 years old suffer from osteoarthritis {Asrani, 2019 #61;Kloppenburg, 2020 #63;Moon, 2020 #62}. Chronic diseases result from metabolic, genetic, inflammatory or environmental causes and most of the time these causes interconnect. Approximately 2 billion adults are obese or overweight laying the path for adipocytes and hepatocytes disfunctions and increased exposition to inflammatory mediators as well {Asrani, 2019 #61}.”

Methods: The material and methods section are not very clear, so it could be structured differently. Sometimes they are a little redundant.

We agree that some sentences may be redundant especially for cell culture description. Our first objective was to give as much information as possible for readers and to avoid them to have to go back and forward to get all the details. In order to make this section more in frame with the chronological process and the result section, we re-structured it and we rephrased a few sentences.

Results and Discussion: I suggest to emphasize the results and discussion of the analysis in the light of the results obtained in the established scientific literature.

We thank you for this comment dedicated to improve manuscript flow and soundness. We tried, at first round, to provide as many relevant references as possible to critically assessed our conclusions. According to you comment we checked for literature update and we added 2 very recent articles to better emphasize the relationships between the three cell types, the metabolic and inflammatory origin of the disorders and the nutritional role of ALE. Therefore, we changed a few sentences in this section and implemented the discussion with the 3 new following paragraphs.

Line 594: “In a very recent paper published in 2021 from Liao et al, authors demonstrated, in a mice model of steatohepatitis and liver damages, that ALE ameliorates both, hepatic oxidative stress and lipid metabolism disorder {Liao, 2021 #64}.”

Line 626: “Interestingly, the ALE-related prevention of hepatic oxidative stress and lipid metabolism disorder observed in a mouse model of steatohepatitis was associated with a lower IL-1β expression that accounts for osteoarthritis onset {Liao, 2021 #64;Wauquier, 2019 #8}. Indeed, the dysmetabolic and inflammatory context responsible for liver and adipose tissues conditions is also the driving force for osteoarthritis establishment. Thus ALE benefits applying to hepatocytes and adipocytes may be extended to chondrocytes.”

Line 648: “Consistently, in 2021, Matsuda and al. found that sesquiterpene lactones from artichoke leaves, including cynaropicrin, inhibit NO production and the induction of iNOS via JAK-STAT and NF-kB in osteoclast precursors RAW264.7 cells {Matsumoto, 2021 #65}.”

Besides, we added a sentence to discuss the NMR data and their relevance regarding the origin of the metabolites. Hopefully, it may contribute to improve the flow between the two paragraphs.

Line 560: “Our clinical ex-vivo approach is integrative and considers the systemic aspect of the digestion, in particular the hepato-intestinal barrier. It allows to test the whole metabolites found in serum after ALE ingestion. Consequently, we keep intact the potential nutritional synergies that can occur at the whole-body level but, it remains difficult with our results to associate an isolated metabolite to a biological effect. Besides, we cannot exclude any contribution of endogenous anti-inflammatory or antioxidant mediators. Nevertheless, we did not observe the appearance of glutathione characteristic signals after ALE ingestion indicating that its concentration is not significantly increased (figure S2). Furthermore, when degradation products of chlorogenic acid were added to a naive serum sample, they led to similar chemical shift variations to those observed for the 100 min serum samples. The parallel between NMR signals strongly suggests that the apparent metabolic changes in the serum profile after ingestion are due to the presence of the ALE polyphenol metabolites or their degradation products. Thus, in the following parts, we compared the benefits that we observed at the cellular level with the literature data regarding the de-scribed effects of the different constituents of the ALE”

In general, check the format of the text.

Regarding the text format, we used the template provided by the journal and we hope the modifications we made, have improved the manuscript’s flow and clarity.

Reviewer 2 Report

In this manuscript, Wauquier et al describe metabolic and anti-inflammatory effects of human enriched serum after artichoke leave extract (ALE) absorption. The authors designed an innovative ex-vivo clinical trial considering the metabolites produced by the digestive tract after artichoke extract ingestion. Human sera, enriched with these metabolites, were collected and incubated with human hepatocytes, human chondrocytes and adipocyted. The results show ALE protects hepatocytes from a lipotoxic stress, prevents adipocytes differentiation and hyperplasia and exert chondroprotective properties.

The manuscript is interesting and technically sound.

You show, in the aromatic region of NMR spectra, molecules that certainly originate from the artichoke in patients’serum , then for sure you show the serum is acting in reducing abovementioned cellular parameters but (1) how can you correlate the aromatic region to this is not clear? (2) Can not be for example vitamin C or an indirect effect of the natural antioxidant system of blood; can it be altered level of glutathione, which is also visible by NMR?

This is mandatory for this reviewer because if these are the results of artichoke consumption instead to take ALE people should only take Vitamin C. In alternative, the authors should clarify this question on discussion section.

Author Response

The manuscript is interesting and technically sound.

Thank you for your comments. We really appreciate that you found our study “interesting and sound” and appropriately designed with conclusions that are supported by our results.

You show, in the aromatic region of NMR spectra, molecules that certainly originate from the artichoke in patients’serum , then for sure you show the serum is acting in reducing abovementioned cellular parameters but (1) how can you correlate the aromatic region to this is not clear? (2) Can not be for example vitamin C or an indirect effect of the natural antioxidant system of blood; can it be altered level of glutathione, which is also visible by NMR?

This is mandatory for this reviewer because if these are the results of artichoke consumption instead to take ALE people should only take Vitamin C. In alternative, the authors should clarify this question on discussion section.

Our clinical ex-vivo approach considers the whole metabolites found in serum after ALE ingestion. Thus, we fully agree that the biological effect observed on cells may be associated with a family of compounds rather than an isolated metabolite. To reach this aim, volunteers were fasted for naïve serum collection and we chose 100min post ALE ingestion for metabolites enriched serum as the result of the maximum variations observed in NMR. The differences observed in cellular parameters result from the comparison between those two sera. The max NMR variations that occurs at 100min corresponds to the phenolic compounds signals consistently with the maximum antioxidant capacity of the serum (100min).

 (1) To clarify this point, we have modified the text to remind that the artichoke leaf extract contains a high amount of polyphenol compounds. We also added a figure in supplementary (Figure S2) material showing that there were minor variations in the aliphatic region.

Line 319: “In agreement with the high level of phenolic compounds in the ingested ALE, the overlay of 1H NMR spectra as a function of time showed that the highest variations were observed in the phenolic compounds’ region and for the 100 min samples (Figure 2a, b and S2).”

(2) The ascorbic acid content in ALE is close to zero. Thus, the observed variations are unlikely due to its presence. Besides, to emphasize that the observed variation in the NMR metabolomic profile cannot not be due to ascorbic acid, a sentence at the beginning of the NMR results was added to remind that trial were conducted on fasted volunteers.

Line 307: “To determine the time frame of the ALE absorption, 3 fasted volunteers were given the extract. Then blood was collected every 20 minutes for 4 hours and the resulted sera were analyzed by 1H NMR spectroscopy. Figure 1 shows representative 1H NMR spectra of basal human serum, as measured by 600 MHz 1H NMR spectroscopy. The spectra display a wide range of metabolites such as phenolic compounds, sugars, amino acids and other metabolites.”

Regarding glutathione or other endogenous anti-inflammatory or antioxidant mediators we cannot exclude an effect. Nevertheless, the low level in the basal serum (10 µM) does not allow its detection by NMR spectroscopy and during our assay, we did not observe the appearance of glutathione characteristic signals after ALE ingestion indicating that its concentration is not significantly increased. According to the literature and in our experimental conditions, the presence of Glutathione would be revealed by the following peaks: 4.202 ppm (q), 3.782 (m), 2.973 (dd), 2.542 (m), 2.155 (m). The absence of the characteristic peaks can be verified in the figure S2. As requested, a sentence was added to the discussion section.

Line 560: “Our clinical ex-vivo approach is integrative and considers the systemic aspect of the digestion, in particular the hepato-intestinal barrier. It allows to test the whole metabolites found in serum after ALE ingestion. Consequently, we keep intact the potential nutritional synergies that can occur at the whole-body level but, it remains difficult with our results to associate an isolated metabolite to a biological effect. Besides, we cannot exclude any contribution of endogenous anti-inflammatory or antioxidant mediators. Nevertheless, we did not observe the appearance of glutathione characteristic signals after ALE ingestion indicating that its concentration is not significantly increased (figure S2). Furthermore, when degradation products of chlorogenic acid were added to a naive serum sample, they led to similar chemical shift variations to those observed for the 100 min serum samples. The parallel between NMR signals strongly suggests that the apparent metabolic changes in the serum profile after ingestion are due to the presence of the ALE polyphenol metabolites or their degradation products. Thus, in the following parts, we compared the benefits that we observed at the cellular level with the literature data regarding the de-scribed effects of the different constituents of the ALE.”

Reviewer 3 Report

dear colleague the manuscript is written and processed well. one of the points that I would have added is the dicersa amount of extract, 8g are really many. usually in our trials we use very low quantities on the 500 mg maximum 1g, but given the results shown there are no toxicological effects even if there is no long-term evaluation. in the article there is no characterization of the extract, I would have expected a chromatographic profile of the extract.

Author Response

dear colleague the manuscript is written and processed well. one of the points that I would have added is the dicersa amount of extract, 8g are really many. usually in our trials we use very low quantities on the 500 mg maximum 1g, but given the results shown there are no toxicological effects even if there is no long-term evaluation. in the article there is no characterization of the extract, I would have expected a chromatographic profile of the extract.

Thank you very much for your compliments.

We fully agree that 8g is quite a lot. As mentioned in the discussion section (lines 662 to 679) this is a single dose exposition that is dedicated to collect a serum enriched with the highest concentration. To set the dose we used a metabolic weight conversion table to transpose from preclinical data and we relied on (1) published clinical chronic exposures (we are 2.5 times higher than the daily dose used for a chronic exposure in Huber’s study (3.2g for 12 weeks)), (2) the recommendation for the polyphenol daily intake that surrounds 1 g (we are 2.2 times higher) and (3) a previous ex vivo clinical trial we conducted with a single dose of 3g of polyphenols originating from olive and grape (we are 27% lower). The reference to this previous ex vivo clinical study was added to the discussion section (line 677).

Regarding the chromatographic profile, we provided it on a new supplementary Figure S1

Round 2

Reviewer 2 Report

The authors have conscientiously addressed the concerns raised in the original review.